# Latent Harmony: Synergistic Unified UHD Image Restoration via Latent Space Regularization and Controllable Refinement

**Yidi Liu**[1], **Xueyang Fu**[1,*], **Jie Huang**[1], **Jie Xiao**[1], **Dong Li**[1],
**Wenlong Zhang**[2,*], **LEI BAI**[2], **Zheng-Jun Zha**[1]
[1]University of Science and Technology of China      [2]Shanghai AI Laboratory
liuyidi2023@mail.ustc.edu.cn,  xyfu@ustc.edu.cn ,[*] Corresponding Author

## Abstract

Ultra-High Definition (UHD) image restoration struggles to balance computational efficiency and detail retention. While Variational Autoencoders (VAEs) offer improved efficiency by operating in the latent space, with the Gaussian variational constraint, this compression preserves semantics but sacrifices critical high-frequency attributes specific to degradation and thus compromises reconstruction fidelity. Consequently, a VAE redesign is imperative to foster a robust semantic representation conducive to generalization and perceptual quality, while simultaneously enabling effective high-frequency information processing crucial for reconstruction fidelity. To address this, we propose *Latent Harmony*, a two-stage framework that reinvigorates VAEs for UHD restoration by concurrently regularizing the latent space and enforcing high-frequency-aware reconstruction constraints. Specifically, Stage One introduces the LH-VAE, which fortifies its latent representation through visual semantic constraints and progressive degradation perturbation for enhanced semantics robustness; meanwhile, it incorporates latent equivariance to bolster its high-frequency reconstruction capabilities. Then, Stage Two facilitates joint training of this refined VAE with a dedicated restoration model. This stage integrates High-Frequency Low-Rank Adaptation (HF-LoRA), featuring two distinct modules: an encoder LoRA, guided by a fidelity-oriented high-frequency alignment loss, tailored for the precise extraction of authentic details from degradation-sensitive high-frequency components; and a decoder LoRA, driven by a perception-oriented loss, designed to synthesize perceptually superior textures. These LoRA modules are meticulously trained via alternating optimization with selective gradient propagation to preserve the integrity of the pre-trained latent structure. This methodology culminates in a flexible fidelity-perception trade-off at inference, managed by an adjustable parameter $\alpha$. Extensive experiments demonstrate that *Latent Harmony* effectively balances perceptual and reconstructive objectives with efficiency, achieving superior restoration performance across diverse UHD and standard-resolution scenarios. The code will be available at https://github.com/lyd-2022/Latent-Harmony.

## 1   Introduction

Image restoration [1–7] aims to recover high-quality images from their low-quality degraded versions with semantic recovery and detail reconstruction, which often struggles to handle unknown corruptions in real-world scenarios. To address this, all-in-one image restoration methods [8–10] develop a single model for multiple degradation types, striving for broad generalization. However, they often face computational efficiency challenges, particularly at high resolutions. In contrast, **Ultra-High Definition (UHD) image restoration** [11–16] specifically targets the immense data scale and intricate detail preservation required for 4K images. To meet complex scenes with various degradations and

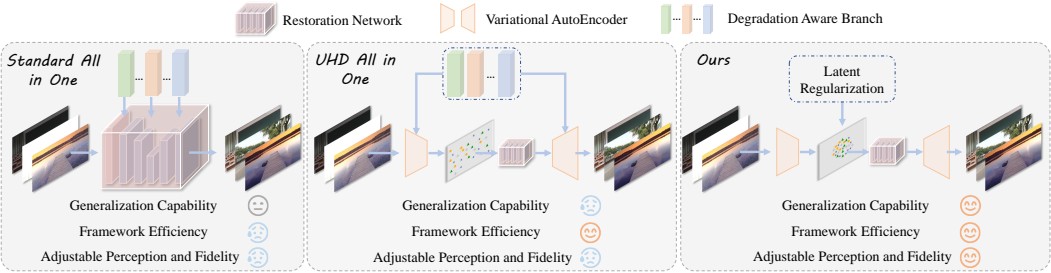

Figure 1: Comparison with existing mainstream methods.Our method outperforms existing standard and UHD all-in-one approaches by leveraging latent regularization, achieving superior efficiency and generalization without requiring degradation-aware branches, while enabling adjustable fidelity and perceptual quality during inference.

high resolution, **UHD all-in-one image restoration** [11] is developed based on the above works, amplifying the challenges of robust generalization, extreme computational efficiency, and meticulous detail fidelity.

To enhance efficiency, existing methods employ Variational Autoencoders (VAEs) to migrate the core restoration process to a lower-dimensional, compact latent space [14, 17, 11, 18]. This approach allows downstream de-degradation networks to operate on significantly smaller latent space, and finally decoding step back to the original space to reconstruct fidelity details while preserving semantics.

However, directly applying VAEs to complex all-in-one UHD image restoration tasks reveals inherent limitations. The VAE's typically achieve compression through Gaussian variational inference, excels at preserving global robust semantic representations [19]. Yet, this mechanism often leads to the loss of critical high-frequency attributes intertwined with degradation characteristics, restricting VAE's ability for high-fidelity reconstruction with the loss of fine details and textures affected by various corruptions. Therefore, a fundamental challenge arises: how to redesign VAE mechanisms to effectively trade-off two crucial properties: (1) extracting degradation-robust semantic representations that ensure good generalization and perceptual quality, and (2) ensuring that these latent representations, upon reconstruction to original pixel space, can adequately process and represent degradation-related high-frequency information for high reconstruction fidelity.

To address this challenge, this paper proposes *Latent Harmony* (LH), a novel two-stage synergistic framework. The core idea of the LH framework is to enable the latent representation to possess both strong semantic robustness and high reconstruction capability by simultaneously regularizing the latent space and imposing high-frequency-related reconstruction constraints.

The first stage introduces the LH-VAE as the foundation for all-in-one UHD image restoration. In its encoding process, building upon the VAE's original Gaussian distribution constraint for latent, the LH-VAE further incorporates progressive degradation perturbation and visual semantic constraints to enhance latent semantic robustness. Concurrently, during its decoding process, latent space equivariance constraints are introduced to improve the latent representation's intrinsic ability to reconstruct high-frequency components. This stage aims to construct a generalized latent space resilient to various degradations and possessing a more balanced frequency characteristic for reconstruction.

The second stage is built on the above VAE. This stage involves joint training with a restoration model, addressing the VAE co-optimization and the perception-fidelity balance. After initially training a latent space restoration network with a fixed LH-VAE, an innovative high-frequency-guided Low-Rank Adaptation (HF-LoRA) fine-tuning mechanism is introduced. To manage degradation-sensitive high-frequency information and enhance fidelity, Fidelity-oriented HF-LoRA (FHF-LoRA) is introduced into the encoder, guided by a high-frequency alignment loss that aligns with the restoration model. Meanwhile, to enhance the perceptual quality of reconstructed output, perception-oriented HF-LoRA (PHF-LoRA) is incorporated into the decoder guided by a high-frequency perception loss. These LoRA modules are trained via alternating optimization with the corresponding losses, thereby protecting the pre-trained VAE's structure from potentially disruptive gradients. Furthermore, the framework allows users to flexibly balance the fidelity-oriented and perception-oriented high-frequency contributions in the final output via an adjustable parameter $\alpha$ during inference.

The main contributions of this paper include:

- We construct a new Latent Harmony two-stage framework, which systematically addresses the multiple trade-off challenges in UHD all-in-one image restoration.

- We design a new latent space regularization strategy that combines progressive degradation, semantic alignment, and equivariance constraints to construct a high-quality generalized VAE latent space.

- We propose a pioneering high-frequency-guided LoRA fine-tuning paradigm that optimizes encoder LoRA for fidelity and decoder LoRA for perception, achieving a synergistic solution for enhanced performance, VAE structural integrity, and controllable output characteristics.

- Extensive experiments demonstrate the superiority of the proposed framework across various UHD and standard-resolution degradation scenarios.

## 2 Related Work

### 2.1 UHD and All-in-One Image Restoration

**Ultra-High Definition (UHD) image restoration** focuses on recovering high-fidelity images from low-quality UHD observations [14–16, 20–22]. The primary obstacles are the prohibitive computational cost of processing high-dimensional data and the critical need to preserve fine-grained, high-frequency details. Directly applying deep models in the pixel space is often computationally infeasible. Consequently, a dominant strategy is the downsample-enhance-upsample paradigm. Methods like UHDFour [23] perform 8x downsampling to enable inference on edge devices, while others like UHDformer [16] and UDR-Mixer [24] use high-resolution features or frequency modulation to guide restoration in a lower-resolution space. However, this downsampling approach inherently discards information crucial for intricate textures in UHD images, imposing an upper bound on the achievable restoration quality. An alternative, efficiency-oriented approach leverages latent space models, such as Variational Autoencoders (VAEs), to perform restoration in a compressed latent domain. For example, DreamUHD [14] employs a VAE with frequency-domain enhancements, and CD²-VAE [18] uses a VAE with feature decoupling to maintain background fidelity while removing degradations. These works underscore the potential of carefully designed latent space models for UHD restoration. Nevertheless, a key challenge remains: enhancing the representational capacity of the latent space for complex degradations while mitigating the intrinsic limitations of VAEs, such as the trade-off between generalization and reconstruction accuracy.

**All-in-One image restoration** aims to devise a unified model capable of addressing diverse, mixed, or unknown degradation types, necessitating exceptional generalization and adaptability [25, 10, 9, 26–28]. Unlike traditional methods that are tailored to specific degradations, All-in-One models are better suited for real-world scenarios with complex degradation patterns. Key challenges include managing degradation heterogeneity, mitigating task conflicts, and achieving awareness of unseen degradations. A prevalent architecture combines a degradation-aware branch with a powerful image restoration backbone. The former often utilizes Mixture-of-Experts (MoE) [29, 28] or Prompting [30–32], while the latter typically adopts architectures like Restormer or NAFNet. For instance, PromptIR [9] introduced a prompt-based mechanism to adapt to various degradations, and MoCE-IR [33] leverages an MoE design for specialized expert handling. Despite their strong performance, these methods often exhibit limited efficiency, which makes full-resolution inference on UHD images computationally demanding on consumer-grade GPUs and thus hinders their practical deployment.

### 2.2 Variational Autoencoders and Latent Space Optimization

**Variational Autoencoders** VAEs are widely used in image restoration due to their encoder-decoder structure, which maps high-dimensional images to a low-dimensional latent space. The VAE objective, the Evidence Lower Bound (ELBO), creates a fundamental tension between reconstruction fidelity and latent space regularity, the latter enforced by a KL divergence term that constrains the latent posterior to a prior distribution. This trade-off is critical: excessive regularization can lead to blurry reconstructions from an information-starved latent space, whereas prioritizing fidelity can weaken the latent structure and impair generalization. While VAEs are foundational to tasks like denoising and super-resolution and form the backbone of Latent Diffusion Models (LDMs), their inherent information compression often leads to the loss of high-frequency details—a significant drawback for UHD restoration. Recent works aim to mitigate this; for example, FA-VAE [34] uses frequency-

complementary modules, and Wavelet-VAE [35] and LiteVAE [36] leverage wavelet transforms to better preserve high-frequency components.

**latent space regularization** To overcome the limitations of standard VAEs, researchers have developed advanced strategies for latent space regularization. Beyond simply tuning the KL divergence weight as in $\beta$-VAE [37], these include applying contrastive learning for improved discriminability (e.g., Hi-CDL in CD²-VAE [18]), imposing geometric constraints on the latent manifold [38, 39], and employing diffusion-based decoders to enhance generation quality (e.g., $\epsilon$-VAE [40]). Such techniques aim to make latent representations more robust to transformations like image degradations. For instance, aligning VAE latents with features from powerful pre-trained vision models like DINOv2 [41], as done in VAVAE [42], injects valuable semantic priors that improve robustness. Furthermore, methods like REPA [43] and REPA-E [44] explore feature alignment losses to optimize VAE and LDM co-training, which also refines the latent structure. These advancements suggest that integrating multiple regularization strategies—particularly by combining external priors with internal structural constraints—is a promising direction for creating a latent space optimized for All-in-One UHD image restoration. This principle forms the foundation of our proposed Latent Harmony framework.

## 3 Motivation

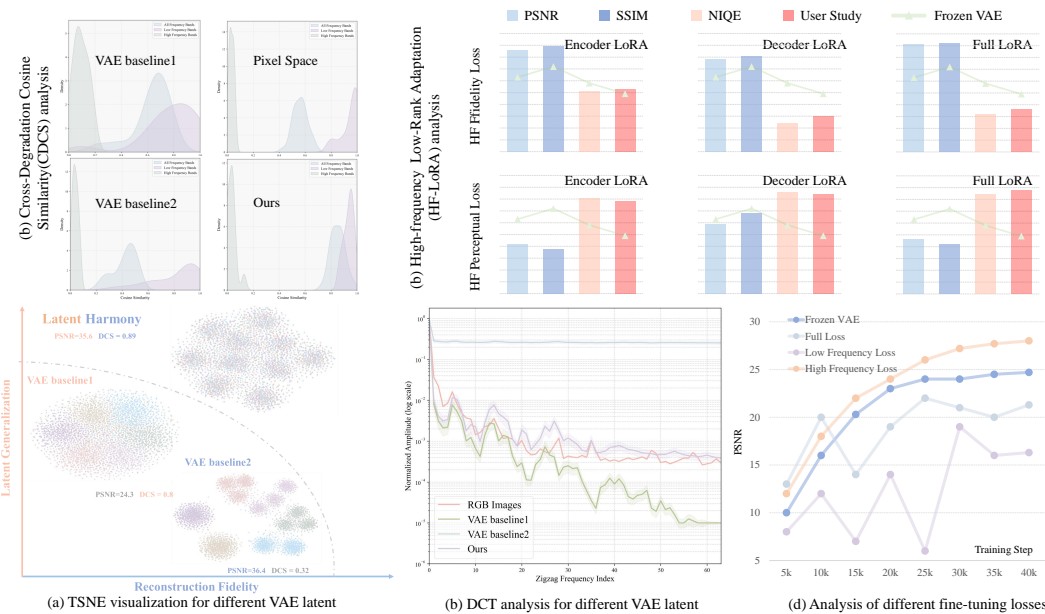

Figure 2: Motivation Analysis. (a) t-SNE visualization of VAE latents under diverse degradations, showing Baseline2's degradation-sensitive clustering versus our method's semantic clustering. (b) Cross-degradation cosine similarity (CDCS) analysis, with higher CDCS in high-frequency bands. (c) DCT spectral analysis, revealing Baseline1's low high-frequency components and Baseline2's elevated components, indicating a reconstruction-generalization trade-off via latent high-frequency proportions. (d) Fine-tuning loss comparison, highlighting stable downstream gains with high-frequency loss. (e) HF-LoRA experiments, demonstrating optimal fidelity and perceptual gains from encoder (fidelity loss) and decoder (perceptual loss) fine-tuning.(Note: All metrics in (e) are normalized to a positive scale, where higher values indicate better performance)

### 3.1 Latent Space Representation: Generalization vs. Reconstruction

To inform our VAE's design for UHD all-in-one image restoration, we conducted experiments comparing two VAE baselines: standard VAE (baseline1) and VAE with enhanced reconstruction (baseline2, Appendix). By analyzing the Cross-Degradation Cosine Similarity (CDCS) of their latent representations for diverse degraded inputs in Fig. 2(b), we find that stronger reconstruction

(baseline2) yields lower latent CDCS (i.e., more diverse among degradations), even below the input's pixel-space CDCS. The t-SNE in Fig. 2(a) depicts a similar phenomenon that stronger reconstruction promotes degradation-driven clustering in latent space, undermining content-based organization. **These results suggest that enhancing VAE's reconstruction capability makes the latent space more sensitive to input degradations**, posing challenges for downstream restoration networks.

We then further perform frequency-domain analysis for latent representations in Fig. 2(b), depicting that high-frequency exhibits low CDCS and low-frequency depicts higher CDCS. Subsequently, a more detailed frequency analysis is performed for both pixel and latent spaces, referring to [45]. Results in Fig. Figure 2(c) indicate that VAEs with stronger reconstruction capabilities (baseline2) tend to encode a significantly higher proportion of high-frequency components in their latent space compared to the pixel space. **This implies that a VAE with strong reconstruction ability encodes ample high-frequency information to manage the challenging task of decoding consistent fine details.**

Moreover, we find that the high-frequency components inherently exhibit low CDCS (Fig. 2(b)) among degradations in latent. Based on the above observations, **an excessive proportion of high frequencies is beneficial for detailed reconstruction but compromises the latent space's robustness to varied degradations.** Therefore, our **first core impetus** is to constrain the latent space to a more moderate high-frequency proportion, balancing latent representation robust to degradations (high CDCS in latent) for generalization and has good reconstruction capabilities (low CDCS in latent).

## 3.2 VAE Co-optimization: Downstream Adaptation vs. Structural Preservation

Following the pre-trained VAE with initial generalization capabilities, a critical question arises: should this VAE remain fixed for downstream restoration tasks, or can downstream restoration supervision signals further adapt the pre-trained VAE to improve the final restoration performance?

While co-optimizing the VAE with the downstream restoration network theoretically promises improved overall performance, direct joint optimization is fraught with risks. Drawing parallels from Latent Diffusion Models (LDMs), where directly applying the main diffusion loss to the VAE can be detrimental [44]. We conceptualize an observational experiment, the pretrained VAE is fine-tuned by backpropagating the restoration loss $L_{Res}$ from the downstream network $R_\theta$. As illustrated in Fig. 2(d), unfreezing the VAE yields faster initial PSNR gains compared to its frozen version; however, continued training often results in performance oscillations. This phenomenon can be attributed to the direct optimization pressure of $L_{Res}$, which compels the encoder to prematurely and aggressively remove degradations from the input. Consequently, **this direct "encoder-latent-restoration-decoder" optimized paradigm tends to devolve into a simplified bottleneck structure geared toward direct pixel-level restoration, thereby disrupting the learned latent space structure.** In contrast, **while a frozen VAE ensures training stability, its lack of adaptation to the restoration task leads to a performance bottleneck that cannot be overcome in later training stages.**

The success of REPA-E [44] achieves effective co-optimization using representation alignment, suggesting a path. In our restoration task, while Stage 1 for VAE's generalization might reduce some high-frequency components, the final reconstruction quality depends on recovering these details. This motivates using high-frequency information alignment as a "bridge" loss for co-optimization.

In our restoration task, as discussed in Section 3.1, enhanced generalization often sacrifices some high-frequency information, while low-frequency components are effectively encoded within a structured latent space. **To balance the preservation of the pre-trained latent structure in VAE with performance gains in downstream restoration tasks during joint optimization, we focus the optimization objective on high-frequency components, employing high-frequency information alignment as a "bridge" loss for co-optimization.** As shown in Fig. 2(d), backpropagating the high-frequency loss to update the VAE maintains training stability while overcoming performance bottlenecks of restoration. Conversely, using a low-frequency alignment loss leads to training instability. Thus, our **second core impetus** is to introduce high-frequency alignment loss as a bridge for joint optimizing VAE and restoration network, preserving the VAE's pre-trained, highly generalizable representations while achieving further performance improvements driven by downstream tasks.

### 3.3 High-Frequency Restoration From Latent: Perception vs. Fidelity

To address the need for task-specific VAE adaptation through high-frequency guidance (see Section 3.2), we focus on designing the high-frequency alignment loss and analyzing its impact on output quality. High-frequency detail recovery entails a trade-off between perceptual quality and fidelity. Pixel-level losses ($L_{pix}$), typically formulated as Maximum Likelihood Estimation, minimize both Systematic Effect (SE) that affects fidelity via regressable components like edges, and Variance Effect (VE) that influences perception through non-regressable textures [46]. Minimizing $L_{pix}$, including its high-frequency component, suppresses SE and VE, yielding high PSNR but perceptually flat outputs due to reduced VE. This is expressed as:

$$\min_{\hat{y}} \left\{ \underbrace{\mathbb{E}_y \left[ \mathcal{L}(y, \mu_{\hat{y}}) - \mathcal{L}(y, \mu_y) \right]}_{\text{SE: LF + regressable HF}} + \underbrace{\mathbb{E}_{y,\hat{y}} \left[ \mathcal{L}(y, \hat{y}) - \mathcal{L}(y, \mu_{\hat{y}}) \right]}_{\text{VE: non-regressable HF}} \right\}, \tag{1}$$

where $\mathcal{L}$ is a symmetric loss, $y \sim p(y|x)$, $\hat{y}$ estimates $y$, and $\mu_y$, $\mu_{\hat{y}}$ are their respective means.

**(a) Fidelity-Oriented High-Frequency Restoration:** This approach prioritizes the faithful extraction or disentanglement of authentic high-frequency components from the input signal, aligning with the ground truth $I_{\text{clean}}$. It emphasizes the "traceability" of high-frequency details, aiming to closely match the ground truth and achieve high fidelity metrics. However, its efficacy is constrained by the availability of residual high-frequency information in the input. Moreover, the suppression of variance effects (VE) can result in monotonous textures, thereby limiting overall perceptual quality.

**(b) Perception-Oriented High-Frequency Generation:** This strategy focuses on generating visually natural high-frequency details, which may not precisely map to the input signal but rely heavily on learned priors about natural images' appearance high frequencies. It prioritizes the visual "plausibility" of high-frequency information, aiming to preserve or shape VE for enhanced visual quality. However, it may introduce structural inaccuracies or hallucinations and compromise fidelity.

The analysis reveals that mechanisms targeting structural fidelity (SE reduction) and texture perception (VE preservation/shaping) inherently pursue distinct optimization objectives. To address these coupled objectives, we introduce two independent Low-Rank Adaptation (LoRA) modules. Specifically, we fine-tune the VAE's encoder, decoder, or both using fidelity-oriented and perception-oriented high-frequency losses, respectively, and evaluate their impact on fidelity and perceptual metrics. As shown in Fig. 2(e), **fine-tuning the VAE's encoder with the fidelity loss enhances fidelity metrics with minimal perceptual quality degradation, while fine-tuning the VAE's decoder with the perceptual loss improves perceptual metrics at a modest cost to fidelity.**

This insight forms **our third core impetus**: within a high-frequency-alignment-based VAE fine-tuning framework, we propose differentiated mechanisms. One mechanism focuses on faithfully extracting and aligning regressable high-frequency components to enhance fidelity in VAE's encoder, while the other concentrates on generating perceptually superior non-regressable high-frequency details to improve perceptual quality in VAE's decoder. We posit that balancing these independently guided mechanisms can effectively synergize fidelity and perception.

## 4 Methodology

Building on Section 3, this chapter details our proposed Latent Harmony two-stage synergistic framework. This framework, through latent space regularization in the first stage and high-frequency guided controllable refinement in the second stage, addresses the trade-offs between: (1) latent space generalization and reconstruction fidelity, (2) VAE co-optimization with downstream tasks versus structural preservation, and (3) the final output's perceptual quality versus fidelity.

### 4.1 Stage One: Constructing a Generalizable Latent Space Representation

The objective of this stage is to train a Variational Autoencoder (VAE) comprising an encoder $E_\phi$ and a decoder $D_\psi$, such that it learns a latent space $Z$ robust to various degradations. The base training follows the standard VAE objective, optimizing a loss $L_{VAE}$ that includes an L1 reconstruction loss on clean images $I_{clean}$ and a KL divergence regularizer:

$$L_{VAE} = \|D_\psi(E_\phi(I_{clean})) - I_{clean}\|_1 + \lambda_{KL} \cdot \text{KL} \left[ q_\phi(z \mid I_{clean}) \parallel p(z) \right] \tag{2}$$

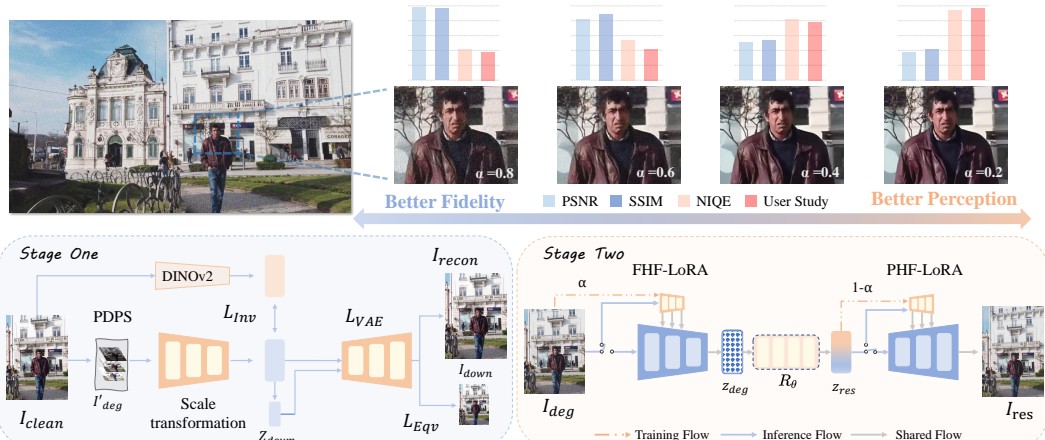

Figure 3: Framework Overview. Stage 1: LH-VAE training employs progressive degradation perturbation, degradation-invariant visual semantic loss $L_{INV}$, and latent space equivariance loss $L_{Eqv}$ to construct a robust, generalizable latent space. Stage 2: Latent space restoration leverages $R_\theta$ and high-frequency-guided LoRA fine-tuning, with Fidelity-oriented HF-LoRA (FHF-LoRA) for the encoder and Perception-oriented HF-LoRA (PHF-LoRA) for the decoder, enabling adjustable fidelity and perceptual quality via parameter $\alpha$ during inference. Results of $\alpha$ tuning are shown in the upper panel, with metrics normalized positively, where higher values indicate better performance.

To counteract the standard VAE latent space's sensitivity to degradations, particularly in high-frequency components , we introduce a progressive degradation perturbation strategy(PDPS). During training, increasingly severe degradations are applied to $I_{\text{clean}}$ over time $t$. This perturbation is probabilistic and can take one of three forms: no perturbation, synthetic degradation, or interpolation with a paired real degraded image $I_{\text{deg}}$. The severity of synthetic degradations is controlled by an increasing function $\text{sev}(t)$, and the interpolation with $I_{\text{deg}}$ is controlled by an increasing coefficient $\beta(t)$. Formally, the perturbed image $I'_{\text{deg}}$ is generated as:

$$I'_{\text{deg}} = \begin{cases} I_{\text{clean}} & \text{with probability } p_0 \\ \text{SynthDeg}(I_{\text{clean}}, \text{sev}(t)) & \text{with probability } p_1 \\ (1 - \beta(t))I_{\text{clean}} + \beta(t)I_{\text{deg}} & \text{with probability } p_2 \end{cases} \tag{3}$$

where $p_0 + p_1 + p_2 = 1$ . $\text{SynthDeg}(I, \text{sev}(t))$ applies a randomly selected set of synthetic degradations (e.g., Gaussian noise, blur, JPEG compression) to image $I$, with their severity controlled by $\text{sev}(t)$, a monotonically increasing function of $t$. The interpolation coefficient $\beta(t) \in [0, 1]$ is also a monotonically increasing function of $t$, signifying a progressively stronger influence of the paired degraded image. This progressive approach ensures learning stability.

On this basis, two key regularization losses are incorporated. The degradation invariance visual semantic loss $L_{INV}$ leverages semantic features $f_{VFM} = \text{VFM}(I_{clean})$ extracted from a pre-trained DINOv2 [41] model as a reference, enforcing the encoder $E_\phi$ to align the encoding $z'_{deg} = E_\phi(I'_{deg})$ of the perturbed image with this reference, learning a degradation-invariant content representation:

$$L_{Inv} = d(z'_{deg}, f_{VFM}) \tag{4}$$

where $d(\cdot, \cdot)$ denotes a distance metric in the feature space. Additionally, the latent space equivariance loss $L_{Eqv}$ constrains the consistency between the decoded result of a randomly downsampled latent encoding $z_{down} = \text{Down}_s(z_{clean})$ and the corresponding downsampled image $I_{down} = \text{Down}_s(I_{clean})$, enhancing scale robustness and reducing reliance on high-frequency components:

$$L_{Eqv} = \|D_\psi(z_{down}) - I_{down}\|_1 \tag{5}$$

The joint optimization objective for this stage combines these terms as:

$$L_{Stage1} = L_{VAE} + \lambda_{Inv}L_{Inv} + \lambda_{Eqv}L_{Eqv} \tag{6}$$

Optimizing this objective yields VAE parameters $(\phi^*, \psi^*)$ that define a latent space exhibiting stronger cross-degradation consistency and more balanced frequency characteristics, establishing

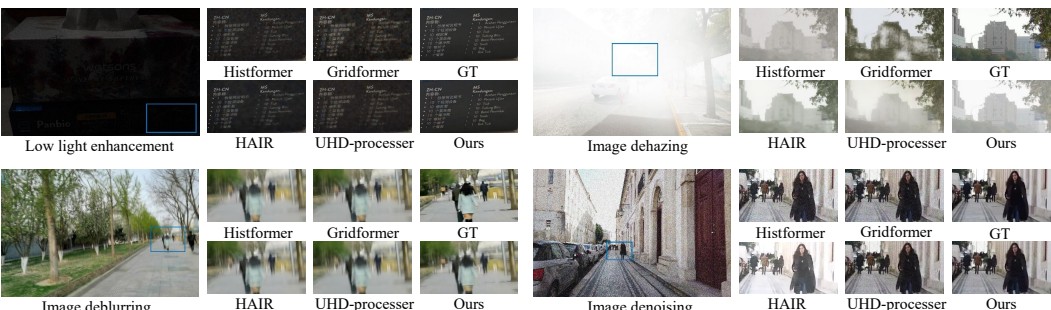

Figure 4: Visual results for four types of degradation removal with other all-in-one methods.

a generalizable foundation for subsequent restoration, albeit potentially at the cost of reducing high-frequency information useful for reconstruction.

## 4.2 Stage Two: High-Frequency Guided Controllable Low-Rank Adaptation

This stage aims to achieve high-quality image restoration using the generalizable latent space $(\phi^*, \psi^*)$ from Stage One, while compensating for lost high-frequency details and providing controllability over the final output. Initially, a latent space restoration network $R_\theta$ is introduced, which processes the encoded degraded latent $z_{deg} = E_{\phi^*}(I_{deg})$ to predict a restored latent $z_{res} = R_\theta(z_{deg})$. This network is trained solely with a standard restoration loss, keeping VAE's parameters $(\phi^*, \psi^*)$ frozen:

$$L_{Res} = \|D_{\psi^*}(z_{res}) - I_{clean}\|_1 \tag{7}$$

Gradients update only the parameters $\theta$ via $\theta \leftarrow \theta - \eta \nabla_\theta L_{Res}$.

Subsequently, to finely restore high-frequency information without compromising the acquired generalization, high-frequency-guided Low-Rank Adaptation (HF-LoRA) fine-tuning is applied to the pre-trained VAE. Low-rank updates $\Delta\phi_{LoRA}$ and $\Delta\psi_{LoRA}$ are introduced to the base parameters $\phi^*$ and $\psi^*$, such that $\phi = \phi^* + \Delta\phi_{LoRA}$ and $\psi = \psi^* + \Delta\psi_{LoRA}$. The LoRA parameters $\theta_{LoRA} = \{\Delta\phi_{LoRA}, \Delta\psi_{LoRA}\}$ are optimized solely by a specific high-frequency alignment loss $L_{HF}$, decoupled from the main restoration loss $L_{Res}$, to preserve the latent space structure learned in Stage One. We design Fidelity-oriented HF-LoRA (FHF-LoRA) for the encoder, guided by a high-frequency alignment loss to enhance fidelity, and perception-oriented HF-LoRA (PHF-LoRA) for the decoder, guided by a high-frequency perception loss to improve perceptual quality, with both modules trained using an alternating optimization strategy.

When optimizing the FHF-LoRA ($\Delta\phi_{LoRA}$), the decoder uses its frozen base parameters $\psi^*$, with the objective being a high-frequency fidelity loss $L_{HF_{Fid}}$ that extracts high-frequency structures from the degraded input consistent with the ground truth:

$$L_{HF_{Fid}} = \|\text{HF}(D_{\psi^*}(E_{\phi^* + \Delta\phi_{LoRA}}(I_{deg}))) - \text{HF}(I_{clean})\|_1 \tag{8}$$

When optimizing the PHF-LoRA ($\Delta\psi_{LoRA}$), the encoder uses its frozen base parameters $\phi^*$, with the objective being a high-frequency perceptual loss $L_{HF_{Perc}}$ to generate visually natural and sharp high-frequency textures, enhancing perceptual quality. This loss is implemented as a GAN-based loss $L_{HF_{GAN}}$, minimizing the adversarial loss for the generator (decoder LoRA) to deceive a high-frequency discriminator $D_{HF}$. The discriminator $D_{HF}$ is optimized by adversarial loss:

$$L_{HF_{GAN}} = -\mathbb{E}_{I_{deg}}\left[\log D_{HF}\left(\text{HF}\left(D_{\psi^* + \Delta\psi_{LoRA}}(R_\theta(E_{\phi^*}(I_{deg})))\right)\right)\right] \tag{9}$$

## 4.3 Inference-Time Control

The differentiated LoRA modules, trained with distinct losses and alternating optimization—$\Delta\phi_{LoRA}$ for fidelity extraction and $\Delta\psi_{LoRA}$ for perceptual generation—provide flexibility during inference. Users can introduce a control parameter $\alpha \in [0, 1]$ to dynamically adjust the contributions of the encoder and decoder LoRA modules to the final result, for instance, via $\phi = \phi^* + \alpha\Delta\phi_{LoRA}$ and $\psi = \psi^* + (1-\alpha)\Delta\psi_{LoRA}$. This mechanism enables a tailored trade-off between maximizing fidelity and optimizing perceptual quality, depending on application requirements.

Table 1: *Comparison to state-of-the-art on four degradations.* PSNR (dB, ↑), SSIM (↑), and LPIPS (↓), and FS represents full-size 4K image inference. FLOPs are computed for an input size of 256×256. **Best** and second best performances are highlighted.

| Method | FS | FLOPs | Params. | Low Light UHD-LL | | Deblurring UHD-blur | | Dehazing UHD-haze | | Denoising UHDN$_{\sigma=15}$ | | UHDN$_{\sigma=25}$ | | UHDN$_{\sigma=50}$ | | Average | | |
|---|---|---|---|---|---|---|---|---|---|---|---|---|---|---|---|---|---|---|
| AIRNet [25] | ✗ | 301G | 9M | 19.24 | .809 | 21.89 | .757 | 18.37 | .812 | 21.33 | .887 | 20.78 | .784 | 18.79 | .475 | 20.07 | .754 | .2843 |
| IDR [47] | ✗ | 88G | 15.3M | 23.12 | .910 | 24.67 | .793 | 19.12 | .768 | 27.48 | .912 | 25.86 | .872 | 24.57 | .654 | 24.14 | .822 | .2684 |
| PromptIR [9] | ✗ | 158G | 33M | 23.44 | .902 | 25.77 | .782 | 19.97 | .727 | 28.43 | .924 | 26.74 | .898 | 23.72 | .584 | 24.68 | .803 | .2571 |
| CAPTNet [30] | ✗ | 25G | 24.3M | 23.96 | .920 | 26.11 | .798 | 19.46 | .868 | 25.58 | .865 | 23.24 | .884 | 21.98 | .508 | 23.39 | .809 | .3466 |
| NDR-Restore [27] | ✗ | 196G | 36.9M | 23.84 | .894 | 24.25 | .802 | 20.08 | .892 | 25.62 | .912 | 24.37 | .897 | 22.94 | .669 | 23.52 | .846 | .3126 |
| Gridformer [48] | ✗ | 367G | 34M | 23.12 | .898 | 25.82 | .783 | 19.24 | .869 | 36.04 | .937 | 31.72 | .898 | 26.24 | .623 | 27.03 | .836 | .3754 |
| DiffUIR-L [49] | ✗ | 10G | 36.2M | 21.56 | .812 | 23.85 | .743 | 18.28 | .864 | 36.84 | .938 | 32.42 | .897 | 26.08 | .648 | 26.51 | .818 | .2564 |
| Histoformer [50] | ✗ | 91G | 16.6M | 23.22 | .908 | 25.62 | .782 | 19.78 | .903 | 26.88 | .845 | 25.64 | .874 | 23.13 | .659 | 24.04 | .829 | .3524 |
| adaIR [10] | ✗ | 147G | 28.7M | 23.57 | .916 | 26.35 | .801 | 18.44 | .901 | 32.84 | .921 | 30.48 | .901 | 26.48 | .672 | 26.36 | .857 | .3429 |
| HAIR [8] | ✗ | 41G | 29M | 25.75 | .922 | 25.78 | .798 | 20.00 | .894 | 35.54 | .916 | 30.84 | .878 | 26.26 | .657 | 27.36 | .847 | .2822 |
| UHDprocesser [11] | ✓ | 4G | 1.6M | 27.11 | .925 | 26.48 | .803 | 20.94 | .923 | 38.94 | .975 | 33.99 | .903 | 27.95 | .677 | 29.23 | .868 | .2541 |
| Ours | ✓ | 3.6G | 1.2M | 27.32 | .926 | 26.98 | .811 | 21.21 | .924 | 39.21 | .978 | 34.78 | .918 | 28.72 | .707 | 29.70 | .877 | .2502 |

Table 2: *Comparison to state-of-the-art on six degradations.* PSNR (dB, ↑), SSIM (↑), LPIPS (↓) and FS represents full-size 4K image inference. FLOPs are computed for an input size of 256×256. **Best** and second best performances are highlighted.

| Method | FS | FLOPs | Params. | Low Light UHD-LL | | Deblurring UHD-blur | | Dehazing UHD-haze | | Denoising UHDN$_{\sigma=50}$ | | Deraining UHD-rain | | Desnowing UHD-snow | | Average | | |
|---|---|---|---|---|---|---|---|---|---|---|---|---|---|---|---|---|---|---|
| AIRNet [25] | ✗ | 301G | 9M | 22.68 | .887 | 23.52 | .876 | 18.24 | .846 | 22.38 | .876 | 26.35 | .876 | 27.38 | .924 | 23.43 | .874 | .1861 |
| IDR [47] | ✗ | 88G | 15.3M | 24.33 | .915 | 25.64 | .788 | 18.68 | .879 | 29.64 | .906 | 28.82 | .906 | 30.48 | .945 | 26.27 | .890 | .1912 |
| PromptIR [9] | ✗ | 158G | 33M | 23.3 | .911 | 26.48 | .805 | 20.14 | .901 | 24.88 | .835 | 28.89 | .897 | 30.78 | .966 | 25.74 | .886 | .2155 |
| CAPTNet [30] | ✗ | 25G | 24.3M | 24.97 | .921 | 26.32 | .796 | 20.32 | .903 | 21.64 | .569 | 29.34 | .908 | 32.21 | .974 | 25.80 | .845 | .2861 |
| NDR-Restore [27] | ✗ | 196G | 36.9M | 25.12 | .885 | 25.64 | .791 | 19.21 | .896 | 31.44 | .915 | 29.24 | .897 | 28.41 | .948 | 26.51 | .889 | .3108 |
| Gridformer [48] | ✗ | 367G | 34M | 23.92 | .898 | 25.68 | .782 | 18.87 | .889 | 32.86 | .915 | 29.37 | .904 | 28.24 | .942 | 26.49 | .895 | .2321 |
| DiffUIR-L [49] | ✗ | 10G | 36.2M | 22.64 | .902 | 25.08 | .785 | 18.62 | .889 | 33.25 | .928 | 27.89 | .886 | 27.36 | .945 | 25.81 | .889 | .1844 |
| Histoformer [50] | ✗ | 91G | 16.6M | 25.73 | .915 | 26.55 | .796 | 18.73 | .897 | 33.05 | .924 | 27.96 | .884 | 27.56 | .971 | 26.59 | .898 | .1855 |
| adaIR [10] | ✗ | 147G | 28.7M | 23.84 | .918 | 26.86 | .803 | 19.34 | .910 | 32.46 | .923 | 28.18 | .901 | 27.72 | .953 | 26.40 | .901 | .2492 |
| HAIR [8] | ✗ | 41G | 29M | 25.22 | .897 | 24.77 | .799 | 18.75 | .883 | 32.50 | .915 | 28.76 | .893 | 27.89 | .968 | 26.31 | .892 | .2607 |
| UHDprocesser [11] | ✓ | 4G | 1.6M | 26.91 | .924 | 26.95 | .807 | 21.81 | .931 | 33.73 | .934 | 29.90 | .915 | 32.73 | .979 | 28.67 | .915 | .1839 |
| Ours | ✓ | 3.6G | 1.2M | 27.14 | .925 | 27.21 | .815 | 22.32 | .936 | 34.17 | .942 | 31.41 | .919 | 33.24 | .982 | 29.24 | .920 | .1822 |

# 5 Experiments

## 5.1 All-in-One and Single-Task Restoration on UHD Scenes

We evaluated the efficacy of our proposed method for the UHD all-in-one restoration task across two experimental configurations: four-degradation and six-degradation settings. As reported in Tables 1 and 2, our approach consistently achieved state-of-the-art performance in both settings while maintaining optimal computational efficiency. Fig. 4 shows visual results of the four-degradation setting, depicting that our method effectively removes degradations while preserving intricate background textures. Although our method is designed for all-in-one tasks, it does not significantly compromise single-task restoration performance, as detailed in the supplementary.

## 5.2 Adaptability and Application Exploration on Standard-Resolution Scenes

Our method employs a unified processing strategy for all degradations, eschewing specialized degradation-aware branches, thereby achieving superior generalization compared to traditional approaches. We validated this generalization capability under two experimental settings: unseen degradations excluded from training and novel composite degradations formed by combining trained degradation types. As shown in Table 4, our approach significantly enhances generalization per-

Table 3: Adaptability in Standard-Resolution Scenarios. Comparisons use LPIPS and FID scores, with lower values indicating superior performance.

| Type | Method | Haze | Rain | Snow | Motion Blur | Raindrop | Low-light |
|---|---|---|---|---|---|---|---|
| Discriminative-based | PromptIR [9] | 0.309/141.05 | 0.097/32.61 | 0.100/18.34 | 0.163/35.79 | 0.189/84.48 | 0.421/189.87 |
| | PromptIR /w Ours | 0.224/121.12 | 0.086/28.68 | 0.092/17.12 | 0.161/35.12 | 0.182/76.84 | 0.378/172.59 |
| LDM-based | Diff-Plugin [51] | 0.340/143.66 | 0.165/39.71 | 0.178/18.08 | 0.147/37.68 | 0.185/60.64 | 0.466/167.63 |
| | Diff-Plugin/w Ours | 0.321/131.12 | 0.162/39.43 | 0.174/18.02 | 0.138/35.42 | 0.146/44.26 | 0.432/152.28 |
| VAE-based | CosAE [17] | 0.328/148.78 | 0.146/38.27 | 0.162/16.78 | 0.186/41.28 | 0.182/49.27 | 0.482/182.24 |
| | CosAE/w Ours | 0.224/128.12 | 0.098/28.79 | 0.121/11.56 | 0.168/36.22 | 0.119/40.62 | 0.382/159.83 |

Table 4: Generalization Verification. PSNR (dB, ↑), SSIM (↑), and LPIPS (↓) are reported.

| Method | Unseen | | | | | | | | | Composite Degradation | | | | | | | | |
|---|---|---|---|---|---|---|---|---|---|---|---|---|---|---|---|---|---|---|
| | UHD-rain | | | UHD-snow | | | UHD-moire | | | LLIE+Noise | | | Haze+LLIE | | | Noise+Blur | | |
| HAIR | 24.32 | 0.924 | 0.392 | 25.43 | 0.903 | 0.223 | 17.28 | 0.798 | 0.446 | 18.12 | 0.812 | 0.492 | 16.72 | 0.862 | 0.439 | 18.42 | 0.854 | 0.471 |
| UHD-processer | 22.72 | 0.924 | 0.342 | 21.82 | 0.918 | 0.267 | 14.32 | 0.778 | 0.489 | 13.28 | 0.842 | 0.428 | 12.38 | 0.872 | 0.462 | 18.28 | 0.824 | 0.492 |
| Ours | **28.13** | **0.892** | **0.233** | **28.92** | **0.967** | **0.184** | **19.26** | **0.898** | **0.326** | **20.33** | **0.882** | **0.342** | **19.82** | **0.904** | **0.328** | **24.28** | **0.898** | **0.278** |

formance in both scenarios, demonstrating that the homogeneous latent space processing paradigm proposed in this work is a more robust alternative to incorporating degradation-aware branches.

The primary objective of this work is to develop a VAE framework tailored for UHD restoration tasks. However, VAEs are also widely employed in standard-resolution scenarios to enhance perceptual quality and reduce the computational demands of diffusion-based restoration methods. To demonstrate the versatility of our approach, we integrated our proposed LH-VAE with three representative standard-resolution restoration methods: discriminative-based, Latent Diffusion Model (LDM)-based, and VAE-based. Experiments were conducted on a multi-degradation benchmark curated from the Gendeg dataset. As shown in Table 3, our method significantly improves the perceptual metrics of all three approaches in standard-resolution settings, thereby validating its generalizability.

## 5.3 Ablation Studies

Table 5: Ablation Studies. Comprehensive ablation experiments validate the efficacy of our approach.

(a) Ablation study of Latent Harmony.

| Configuration | PSNR ↑ | SSIM ↑ | LPIPS ↓ |
|---|---|---|---|
| Latent Harmony | **29.77** | **0.88** | **0.250** |
| w/o $L_{\text{Inv}}$ | 24.28 | 0.79 | 0.292 |
| w/o $L_{\text{Eqv}}$ | 25.68 | 0.82 | 0.302 |
| w/o PDPS | 27.82 | 0.84 | 0.287 |
| w/o FHF-LoRA | 28.12 | 0.86 | 0.286 |
| w/o PHF-LoRA | 29.02 | 0.84 | 0.306 |
| w/o LoRA Fine-Tuning | 28.68 | 0.85 | 0.298 |
| w/o Fine-Tuning | 28.48 | 0.86 | 0.292 |

(b) Inference time comparison.

| DreamUIR | Histformer | UHDprocesser | LH (Ours) |
|---|---|---|---|
| 12.3 | 8.4 | 1.2 | **0.43** |

(c) Ablation Study of Latent Restoration Network.

| metric | Restormer | | NAFNet | | SFHformer | |
|---|---|---|---|---|---|---|
| | Base | +Ours | Base | +Ours | Base | +Ours |
| PSNR (dB) | 24.22 | 29.73/ **+5.51** | 24.63 | 29.68/ **+5.05** | 24.54 | 29.70/ **+5.16** |
| Param (M) | 26.1 | 3.8/ **-85%** | 29.1 | 1.9/ **-93%** | 7.6 | 1.2/ **-84%** |
| FLOPS (G) | 140.9 | 6.2/ **-95%** | 16.1 | 4.7/ **-71%** | 51.0 | 3.6/ **-93%** |
| Runtime (s) | 8.8 | 0.62/ **-92%** | 4.6 | 0.41/ **-92%** | 5.2 | 0.43/ **-92%** |
| FS | ✗ | ✓ | ✗ | ✓ | ✗ | ✓ |

(d) Performance metrics under different $\alpha$ values.

| Metric | $\alpha = 0.2$ | $\alpha = 0.4$ | $\alpha = 0.6$ | $\alpha = 0.8$ |
|---|---|---|---|---|
| PSNR | 28.94 | 29.28 | **29.70** | 29.74 |
| SSIM | 0.862 | 0.867 | **0.877** | 0.878 |
| LPIPS | 0.2218 | 0.2483 | **0.2502** | 0.2904 |
| User | 9.2 | 7.8 | **6.2** | 4.8 |

To validate the contributions of the key components in our "Latent Harmony" framework, we conducted thorough ablation experiments on the UHD all-in-one restoration task, systematically removing or modifying individual components and assessing their impact on performance using PSNR, SSIM, and LPIPS metrics. Results are summarized in Table 5. The ablation of primary components, presented in Table 5(a), confirms the effectiveness of each proposed element. Additionally, the latent space restoration network in Latent Harmony adopts SFHformer [5], and we verify the robustness of our approach across alternative network architectures in Table 5(c). Runtime comparisons, shown in Table 5(b), demonstrate significant efficiency gains over competing methods, underscoring the necessity of eliminating degradation-aware branches. The impact of the tuning parameter $\alpha$ on fidelity and perceptual quality is illustrated in Table 5(d), where increasing $\alpha$ enhances fidelity metrics at the expense of perceptual metrics, validating the tunability of our method's output. Detailed experimental setups, implementation specifics, and additional results and analyses are provided in supplementary.

## 6 Conclusion

This paper addressed critical VAE-based trade-offs in UHD all-in-one image restoration, encompassing latent generalization versus reconstruction, structural integrity during co-optimization, and the perception-fidelity balance. We introduced *Latent Harmony*, a two-stage framework. Its first stage constructs a robust Latent Harmony VAE (LH-VAE) via principled latent space regularization. The second stage features high-frequency-guided LoRA fine-tuning, distinctly optimizing encoder LoRA for fidelity and decoder LoRA for perception, while preserving VAE structure. An inference parameter $\alpha$ enables explicit fidelity-perception control. Extensive experiments validated *Latent Harmony's* superior restoration performance and effective balancing of these trade-offs across diverse scenarios, presenting a promising advancement for UHD image restoration.

# Acknowledgement

This work was supported by the National Natural Science Foundation of China (NSFC) under Grants 62225207, 62436008, 62422609 and 62276243.

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
