# OpenReview forum: "Latent Harmony: Synergistic Unified UHD Image Restoration via Latent Space Regularization and Controllable Refinement"
_NeurIPS.cc/2025/Conference — NeurIPS 2025 poster_

### Official Review · Reviewer_Nq1u · 2025-06-12

**Clarity:** 3
**Significance:** 3
**Originality:** 3
**Rating:** 5
**Confidence:** 5

**Summary:**

The authors propose Latent Harmony, a two-stage VAE-based framework for UHD image restoration, addressing the trade-off between computational efficiency and detail retention. In Stage One, they introduce LH-VAE, which enhances latent space robustness through semantic constraints and degradation perturbations while improving high-frequency reconstruction via latent equivariance. In Stage Two, they integrate HF-LoRA, a High-Frequency Low-Rank Adaptation module with encoder-decoder components optimized for fidelity and perceptual quality, trained via alternating gradient propagation.

**Questions:**

1. Although Figure 1 is presented, it is not explained in the manuscript.
2. Figure 3 shows the entire method overview, but I can't tell what type of degradation the displayed figure shows. Furthermore, the fidelity of the restored images corresponding to different α values shows a positive correlation. Can the author explain this phenomenon?
3. In the first stage of training, the authors propose the PDPS strategy to enhance the robustness of VAE against degradation in the latent space. Among them, p0, p1, and p2 control different perturbations, respectively, but the authors do not provide their values or reasons.
4. Eq 4 and 5 represent two loss functions. However, for Eq 4, the author emphasizes that it is a distance metric, while Eq 5 is an L1 norm. What are the differences between the two equations?
5. In fact, the authors adopt the VAE-based image restoration pipeline, but most of the methods compared in the experiment are based on Transform or diffusion methods. It is necessary to explain the advantages of VAE compared with these methods.

**Ethical Concerns:**

["NO or VERY MINOR ethics concerns only"]

**Final Justification:**

The authors have carefully addressed the issues raised in the previous round of review. They have made effective improvements to the methodology description, experimental design, and results analysis. The revised manuscript has adequately resolved my primary concerns from the initial review—the theoretical logic is now more rigorous, the data support is more robust, and the conclusions are more reliable. Based on these improvements, I recommend accepting the paper in its current form.

**Limitations:**

Yes

**Paper Formatting Concerns:**

The manuscript has no major formatting issues.

**Quality:**

3

**Strengths And Weaknesses:**

Strengths:
1. The motivation is summarized in detail and explained deeply.
2. The experiments are rich and can prove the effectiveness of the proposed method.
Weakness:
1. The authors do not propose an innovative architecture, but only design multiple loss functions and training strategies.
2. The frequent appearance of the "xx's " writing style in the article is not standardized.

---

> ### Author Rebuttal · Authors · 2025-07-31
>
> ### Response to Comments on Architectural Innovation：
> We appreciate the reviewer’s candid feedback. Rather than focusing on designing new architectures, this work prioritizes **problem analysis** and **framework development**. Similar to works such as Noise2Noise[1], RankSR[2], GRIDS[3],  LitCLIP[4] and TVT[5] , our contribution focuses on designing a more effective training strategy and holistic framework tailored to the task’s demands. For the UHD All-in-One (UHD AIO) restoration task, we innovatively unify the computational bottleneck of high-resolution processing and the optimization challenges of multi-degradation scenarios into a **standardized VAE latent space optimization framework**. Through extensive observational experiments, we decompose latent space representation optimization into **three core trade-offs**, leading to the development of a novel, systematic framework and training paradigm. Thus, our primary innovation lies in the **problem identification process** (i.e., the in-depth motivational analysis praised by the reviewer) and the **problem-driven framework design**. Architectural innovation in specific components is not the focus of this work but will be explored in future studies. Moreover, our framework innovation is **orthogonal** to architectural advancements. For instance, integrating our approach with the prompt-learning architecture from UHDProcessor yields further performance improvements, demonstrating that both training framework innovation and network architecture advancements are critical for UHD image restoration.
>
>
> | metric | UHDprocessor | Ours  | UHDprocessor+Ours |
> | ------ | ------------ | ----- | ----------------- |
> | PSNR   | 29.23        | 29.70 | 29.94             |
> | SSIM   | 0.86         | 0.88  | 0.91              |
>
> [1] Noise2Noise: Learning image restoration without clean data [ICML 2018]
>
> [2] RankSRGAN: Generative Adversarial Networks with Ranker for Image Super-Resolution [ICCV 2019 oral]
>
> [3] GRIDS: Grouped Multiple-Degradation Restoration with Image Degradation Similarity [ECCV 2024]
>
> [4] Iterative Prompt Learning for Unsupervised Backlit Image Enhancement [ICCV 2023 oral]
>
> [5] Fine-structure Preserved Real-world Image Super-resolution via Transfer VAE Training [ICCV 2025]
>
> ### Response to Comments on Writing Conventions
> We sincerely thank the reviewer for pointing out the improper use of "xx's" in our manuscript. We acknowledge this oversight and commit to conducting a thorough review and revision of the entire manuscript in the revised version, correcting all non-standard expressions to enhance the professionalism and readability of the paper.
>
> ### Q1: Explanation of Figure 1
> Fig 1 provides a conceptual comparison between our framework and existing mainstream All-in-One image restoration paradigms:
>
> * Left Panel (Restoration Network): Represents traditional methods operating directly in the pixel space, which typically exhibit lower efficiency when processing UHD images.
> * Middle Panel (VAE + Degradation-aware Branch): Illustrates approaches that leverage VAEs for improved efficiency but rely on additional degradation-specific branches or prompts. This increases parameter overhead and limits generalization to unseen degradations.
> * Right Panel (Our Method): Depicts our Latent Harmony framework, which also utilizes VAEs for efficiency but enhances generalization through latent space regularization, eliminating the need for degradation-specific branches. This streamlined design achieves both higher efficiency and stronger generalization.
>
> The figure intuitively conveys our method’s core design objectives across three dimensions—**framework efficiency**, **generalization capability**, and **output controllability**—achieving efficient, universal, and tunable restoration without relying on additional degradation-specific branches. We will include a detailed explanation of Figure 1 in the introduction of the revised manuscript to better contextualize our work.
>
> ### Q2: Details of Figure 3
> Degradation Types in Figure 3:Figure 3 presents an overview featuring a noise degradation sample, with the zoomed-in image highlighting differences from the ground truth (GT). This is merely illustrative; in practice, our training encompasses multiple degradation types simultaneously.
>
> Positive Correlation Between $\alpha$ and Fidelity: The reviewer accurately noted the positive correlation between the $\alpha$ value and fidelity metrics, which is a deliberate aspect of our controllable inference mechanism rather than a coincidental outcome.
>
> The $\alpha$ parameter directly modulates the weight of the fidelity-oriented FHF-LoRA module in the encoder. This module is specifically trained with a high-frequency fidelity loss $L_{HF_{Fid}}$ to reconstruct pixel-aligned details matching the ground truth. Consequently, higher (\alpha) values increase the contribution of the fidelity-driven module, resulting in improved PSNR/SSIM (fidelity metrics). Conversely, lower $\alpha$ values emphasize the perceptual quality-driven decoder LoRA, trained with GAN loss, which generates visually realistic textures with slightly reduced pixel alignment. This trade-off is quantitatively supported by data in Table 5(d) of the main paper. One of our method’s key advantage lies in providing a user-adjustable parameter $\alpha$ , **enabling flexible tuning of the trade-off between fidelity and perceptual quality based on user-specific requirements** .
>
>
>
> ### Q3: Clarification on PDPS Probability Parameters
> We thank the reviewer for this insightful question and apologize for omitting these details in the original manuscript. The specific values and their underlying rationale are as follows.
>
> In our implementation, the probabilities for the Progressive Degradation Perturbation Strategy (PDPS) were set to $p_0=0.6$, $p_1=0.2$, and $p_2=0.2$. The rationale for this distribution is rooted in establishing a strong foundation for reconstruction fidelity while systematically introducing robust generalization capabilities:
>
> * **$p_0=0.6$ (No Perturbation)**: We assign the highest probability to the clean image ($I_{clean}$) to ensure the VAE first learns a stable and high-fidelity representation of the clean data manifold. This serves as a critical anchor, ensuring the model's primary objective remains robust reconstruction before being heavily exposed to diverse degradations.
>
> * **$p_1=0.2$ (Synthetic Degradation)**: A significant portion of the training involves applying various synthetic degradations. This step is the primary mechanism for teaching the model to generalize across a wide range of corruption types and levels in a controlled manne.
>
> * **$p_2=0.2$ (Real Degradation Interpolation)**: We also assign a probability to interpolating with a paired real degraded image ($I_{deg}$). This helps to bridge the domain gap between synthetic corruptions and complex, real-world artifacts, thereby enhancing the model's practical performance and robustness.
>
> We will add these specific values and their detailed rationale to the implementation details in our revised supplementary material.
>
> ###  Q4: Differences Between the Metric Functions in Eq. 4 and Eq. 5
> While both Equations 4 and 5 employ distance metrics, they operate in distinct spaces and serve entirely different objectives.
>
> **Eq. 4:**  $L_{Inv} = d(z_{deg}^{\prime}, f_{VFM})$ . This loss function operates in a high-dimensional semantic feature space, computing the distance between the VAE’s latent encoding $z_{deg}^{\prime}$ and features $f_{VFM}$ extracted by a robust pretrained vision model (DINOv2). Its objective is to ensure semantic consistency, preserving the core semantic representation in the latent space regardless of image degradation. We adopt a cosine similarity-based distance metric $d(\cdot,\cdot)$, as the direction of feature vectors is more critical than their absolute magnitude in high-dimensional feature spaces.
>
> **Eq. 5:**  $L_{Eqv} = |D_{\psi}(z_{down}) - I_{down}|$ .This loss function operates directly in the image pixel space, comparing the decoded image from a downsampled latent encoding $D_{\psi}(z_{down})$ with the directly downsampled original image $I_{down}$. Its goal is to ensure structural equivariance across scales. We use the L1 norm due to its robustness to outliers and its established effectiveness as a standard loss in image reconstruction, compared to the L2 norm.
>
> ### Q5: Comparison of VAE with Transformer/Diffusion Methods
>
> To clarify, our method is not a standalone VAE framework positioned against Transformer or Diffusion models. Instead, it proposes a **“latent space restoration framework”** tailored to UHD image characteristics, in contrast to **traditional “pixel space restoration frameworks”** . This framework’s key advantage lies in shifting computationally intensive restoration tasks from dense pixel space to a compact latent space, significantly improving efficiency, as evidenced in **Table 5c**. Importantly, our approach is **orthogonal** to Transformer and Diffusion methods, as demonstrated by results with Restormer and SFHformer in **Table 5c** and the LDM-based Diff-plugin in **Table 3**. The proposed Latent Harmony VAE framework substantially enhances the **computational efficiency** and **restoration performance** of Transformer and Diffusion methods.

---

### Official Review · Reviewer_kppD · 2025-06-24

**Clarity:** 3
**Significance:** 3
**Originality:** 2
**Rating:** 4
**Confidence:** 3

**Summary:**

This paper proposes Latent Harmony, a two-stage framework for UHD image restoration that enhances the balance between efficiency, reconstruction fidelity, and generalization. In the first stage, a specially designed LH-VAE is trained using progressive degradation perturbation, semantic alignment via DINOv2 features, and latent space equivariance to build a robust and generalizable latent representation. In the second stage, the authors introduce high-frequency-guided low-rank adaptation which fine-tunes the encoder for fidelity and the decoder for perceptual quality using separate high-frequency losses. Extensive experiments show that the method achieves state-of-the-art performance with lower computational cost and strong generalization across various degradation types.

**Questions:**

1. The proposed HF-LoRA tuning strategy uses alternating optimization of encoder (FHF) and decoder (PHF) modules. This is an interesting design, but the supplementary does not report comparisons with joint tuning. It is important to know whether the alternating scheme is empirically superior, or simply a safeguard against interference. Please provide evidence or reasoning for the alternating scheme's effectiveness. For example, does joint tuning degrade latent structure or reduce performance?

2. Several key components—DINOv2 for latent alignment, LoRA for efficient fine-tuning—are derived from prior work. While the integration is novel, it is important to clarify how much of the gain stems from this architecture versus the strength of the DINOv2 prior and the generality of LoRA. Have you conducted experiments with weaker backbones (e.g., vanilla ViT or ConvNeXt) or without the LoRA separation? Additional baselines in the supplementary would help isolate the contribution of the proposed synergy.

3. The progressive degradation perturbation strategy is a central component of Stage 1, yet its exact scheduling policy, degradation diversity, and sensitivity to design choices remain under-explained. While the supplementary material shows some degradation types used, it is unclear how critical the progression (vs. random sampling) is to the learned latent space. Please clarify the rationale behind the PDPS schedule and whether ablation against simpler augmentation strategies (e.g., single-step or random mix) has been performed. This would help disentangle how much benefit comes from PDPS itself versus general data augmentation.

**Ethical Concerns:**

["NO or VERY MINOR ethics concerns only"]

**Final Justification:**

While this work may not introduce novel methodology, it offers a set of useful and well-supported findings regarding the application of existing techniques to UHD image restoration.

**Limitations:**

Yes.

**Paper Formatting Concerns:**

No major formatting violations were observed.

**Quality:**

3

**Strengths And Weaknesses:**

**Strengths**:

* The proposed two-stage pipeline is well-structured and experimentally validated on both UHD and standard-resolution restoration tasks.
* Ablation studies are comprehensive and demonstrate the impact of each component.
* The results show strong quantitative performance with reduced FLOPs and parameter counts, making a compelling case for practical efficiency.

**Weaknesses**:
* Despite empirical gains, the architectural novelty is incremental—main components like VAE regularization, LoRA fine-tuning, and perceptual loss design are all based on existing paradigms.
* The contribution may be limited by reliance on a strong pretraining backbone (DINOv2), which may obscure whether improvements stem from the framework or the visual prior itself.
* Most components (e.g., PDPS, use of DINOv2 for semantic alignment, LoRA for tuning) are adaptations of known techniques rather than fundamental innovations.
* The proposed HF-LoRA modules do not offer architectural novelty beyond standard LoRA tuning, and are primarily differentiated by their loss functions and placement.

---

> ### Author Rebuttal · Authors · 2025-07-30
>
> ### Clarification on Motivation and Novelty
>
> We appreciate the reviewer’s perspective regarding the use of existing techniques such as VAE, LoRA, and DINOv2. We emphasize that our core contribution lies not in inventing these individual components but in providing a novel analytical perspective and a systematic framework for the complex UHD All-in-One (UHD AIO) task through in-depth motivational analysis.
>
> **Novel Analytical Perspective:** For the UHD AIO task, which simultaneously faces **high-resolution computational bottlenecks** and **multi-degradation optimization challenges**, our work introduces a unified approach through the lens of VAE **latent space representation optimization**. Extensive observational experiments in **Fig 2** distill three core trade-offs, forming a novel analytical framework:
> 1. Latent Space Level: Balancing content “generalization” with detail “reconstruction.”
> 2. Optimization Level: Balancing downstream restoration task “fine-tuning adaptability” with VAE prior “structural preservation.”
> 3. Output Level: Balancing decoded image “fidelity” with “perceptual quality.”
>
> **Systematic framework:**
> 1. Addressing Trade-off 1: Based on the insight that “high-frequency encoding impacts generalization” (**Fig 2c**), we design a **novel regularization strategy**. The semantic alignment loss $L_{Inv}$ preserves low-frequency semantics, while the latent equivariance loss $L_{Eqv}$ balances frequency characteristics.
> 2. Addressing Trade-off 2: We introduce a **high-frequency alignment loss** as a bridge for joint optimization across stages, preserving robust low-frequency semantic structures for generalization while precisely compensating suppressed high-frequency details to efficiently adapt to downstream restoration tasks, achieving an optimal balance between structural preservation and performance enhancement.
> 3. Addressing Trade-off 3: We propose a separated target-decoupled LoRA, assigning the conflicting objectives of fidelity and perceptual quality to the encoder and decoder, respectively, optimized alternately. An adjustable parameter $\alpha$ enables flexible trade-offs during inference, allowing users to tailor outputs to specific use cases.
>
> The primary innovation of this work lies in the **comprehensive analysis** of UHD AIO task challenges presented in Fig 2 and the **problem-driven design of a systematic framework**, akin to framework-level innovations in [1-4]. Orthogonal to architectural innovations, our framework, when combined with UHDProcessor, enhances performance, as shown below, demonstrating that both framework and architectural innovations are indispensable to advancing the field.
>
> |metric|UHDprocessor|Ours|UHDprocessor+Ours|
> |-|-|-|-|
> |PSNR|29.23|29.70 |29.94|
> |SSIM|0.86|0.88|0.91|
>
> [1] RankSRGAN: Generative Adversarial Networks with Ranker for Image Super-Resolution [ICCV 2019 oral]
>
> [2] GRIDS: Grouped Multiple-Degradation Restoration with Image Degradation Similarity [ECCV 2024]
>
> [3] Iterative Prompt Learning for Unsupervised Backlit Image Enhancement [ICCV 2023 oral]
>
> [4] Fine-structure Preserved Real-world Image Super-resolution via Transfer VAE Training [ICCV 2025]
>
> ---
>
> ### Q1: Comparison of Alternating Optimization vs. Joint Optimization in HF-LoRA
> We adopt an alternating optimization scheme in HF-LoRA to mitigate gradient conflicts between competing objectives, rather than merely as a precautionary measure.
>
> **Theoretical Rationale:** As outlined in **Eq.1** of main text, our two LoRA modules pursue conflicting objectives. The FHF-LoRA (encoder), guided by a fidelity-driven loss (e.g., L1 loss), favors pixel-aligned, smooth solutions to minimize error. In contrast, the PHF-LoRA (decoder), driven by a perceptual loss (e.g., GAN loss), promotes sharper, sometimes exaggerated textures to enhance visual realism by “fooling” the discriminator. Joint optimization of both LoRA modules aligns gradients for **regressable high-frequency components**(RHF,Eq1) early in training. However, in later stages, conflicting gradients for **non-regressable high-frequency components** (NRHF,Eq1)cause training instability and hinder convergence, limiting benefits to RHF. Our alternating optimization approach mitigates these conflicts, enabling accurate restoration of RHF while allowing FHF-LoRA to smooth NRHF for improved fidelity and PHF-LoRA to generate more realistic NRHF for enhanced perceptual quality.
>
> **Supplementary Experiments:** Comparisons between alternating optimization ($\alpha = 0.6$) and joint optimization, shown below, reveal that joint optimization yields only marginal gains over freezing the VAE in Stage 2, confirming the presence of optimization conflicts. Our alternating optimization strategy overcomes this challenge, achieving substantial improvements in both fidelity and perceptual quality.
>
> |Metric|Frozen VAE|Joint Optimization|Alternating Optimization|
> |-|-|-|-|
> | PSNR ↑  |28.48|28.62|**29.70**|
> | SSIM ↑ |0.86 |0.86| **0.877**|
> | LPIPS ↓|0.292|0.282|**0.250**|
>
> ---
>
> ### Q2: Disentangling Framework Contributions from DINOv2 Prior
>
> The reviewer raises a critical point regarding the need to distinguish our framework’s contributions from those of the powerful DINOv2 prior. We fully agree and have validated this distinction in the submitted supplementary materials.
>
> Comparison with Different Vision Foundation Models: In **Supp Tab 5**, we present a key ablation study comparing performance when using various vision foundation models (MAE, SAM, CLIP, SigLIP) as the backbone for our $L_{Inv}$ semantic loss. Additionally, we incorporated comparisons with the  vanilla ViT and ConvNeXt, as suggested by the reviewer. The complete results are shown below. We selected DINOv2 primarily for its established, **classic status**, yet our framework consistently improves performance across all tested backbones. This strongly demonstrates that the effectiveness of our framework stems from its **semantic alignment design**, rather than solely relying on DINOv2’s capabilities.
>
> |Method|PSNR|SSIM|
> |-|-| -|
> |VIT|29.63|0.870|
> |ConvNeXt|29.59|0.873|
> |MAE|29.62|0.872|
> |SAM|29.66| 0.880|
> |CLIP|29.57| 0.875|
> |SigLIP|29.64| 0.882 |
> | DINOv2 | 29.70 | 0.877|
>
> Effectiveness of Separated LoRA Design: LoRA serves as an efficient fine-tuning mechanism for our high-frequency alignment loss, selected for its established classic, similar to our choice of DINOv2. However, LoRA can be substituted with other parameter-efficient fine-tuning (PEFT) methods, such as Adapter, Visual Prompt Tuning, or DoRA. The results, presented below, demonstrate consistent performance improvements across these PEFT methods, indicating that our method’s gains stem not from LoRA’s efficacy but from the **high-frequency alignment loss strategy** itself. This strategy, derived from observational experiments in **Fig 2d and 2e**, enables **separated fine-tuning** for encoder fidelity and decoder perceptual quality.
>
> | Metric  |VPT | Adapter | DoRA  | LoRA(Ours) |
> |-|-|-|-|-|
> | PSNR ↑|29.66| 29.68 | 29.74 | 29.70|
> | SSIM ↑| 0.89  | 0.86 | 0.87  | 0.88|
> | LPIPS ↓| 0.256 | 0.251| 0.248 | 0.250 |
>
> These results underscore that the **semantic alignment loss** and **efficient joint fine-tuning strategy** in our contribution are independent of specific implementations like DINOv2 or LoRA. Substituting other vision foundation models or efficient fine-tuning methods yields consistent performance improvements.
>
> ---
>
> ### Q3: Design Rationale and Necessity of PDPS
>
> We thank the reviewer for their interest in the Progressive Degradation-aware Pretraining Strategy (PDPS). Below, we elucidate its design rationale.
>
> **Rationale for Progressiveness:** PDPS is grounded in **Curriculum Learning**, following a structured easy-to-hard progression. Our goal is to enhance the VAE’s semantic invariance, enabling it to map diverse degradation types and intensities to a **unified, clean semantic latent space**. As detailed in **Eq 3** of the main paper, PDPS employs probabilities $P_0 = 0.6$, $P_1 = 0.2$, and $P_2 = 0.2$, corresponding to clean images, synthetic degradation perturbations (e.g., JPEG artifacts, low-light, noise, color distortion, blur), and paired degradation image interpolation, respectively. The degradation severity $sev(t)$ and interpolation strength $\beta(t)$ are modeled using a sigmoid-based function, $\frac{1}{1 + e^{-\alpha (t - \tau)}}$ , ensuring a gradual increase in degradation intensity with training step $t$. Training with diverse and severe degradations from the outset (e.g., random mixing) risks convergence failure or learning to “ignore” input signals, hindering content-degradation disentanglement. PDPS starts with simple tasks (clean or lightly degraded images) and progressively increases degradation complexity, providing a stable learning trajectory. This ensures the VAE first establishes a robust foundation for “clean content” before addressing complex degradations.
>
> **Comparison with General Data Augmentation:** Ablation studies in **Tab 5a** of the main paper show that removing PDPS (w/o PDPS) leads to significant performance degradation. We further compare PDPS with two augmentation strategies: (1) random sampling with constant degradation intensity(RSCD) and (2) random sampling with random degradation intensity(RSRD). The results, presented below, indicate that both alternatives disrupt the curriculum learning process, forcing the VAE to learn complex degradations before establishing a stable latent space, resulting in training instability and poor convergence. In contrast, PDPS’s steadily increasing degradation intensity fosters stable training and effectively.
>
> | metric| w/o PDPS|RSCD |RSRD |PDPS|
> |:-:|:-:|:-:|:-:|:-:|
> |  PSNR  |  27.82   | 27.24 | 26.96 | 29.70|
> |  SSIM  |0.84|0.85|0.82|0.88|
>
> In the revision, we will expand on the curriculum learning principles underlying PDPS in the supplementary materials, emphasizing that its progressive design ensures training stability and model robustness.

---

> > ### Comment · Reviewer_kppD · 2025-08-04
> >
> > Thank you for the authors’ response. I appreciate the clarifications and additional results provided in the rebuttal. Given the clarified contributions and empirical findings, I am willing to raise my score. I hope the authors will consider releasing their code in the future to further benefit the community.

---

> > > ### Author Response · Authors · 2025-08-04
> > > **Reply to Reviewer kppD**
> > >
> > > We sincerely appreciate the reviewer's insightful feedback and careful consideration of our rebuttal. We are committed to rigorously incorporating these suggestions in the revision and make the code publicly available upon acceptance.

---

### Official Review · Reviewer_KgHd · 2025-07-01

**Clarity:** 3
**Significance:** 3
**Originality:** 3
**Rating:** 4
**Confidence:** 4

**Summary:**

This paper proposes a high-resolution image restoration method called Latent Harmony. It aims to address a issue: how to ensure computational efficiency while retaining details when inpainting ultra high resolution images such as 4K. To achieve this goal, the paper proposes a two-stage approach to handle the existing defects of VAEs, optimizing the VAEs by introducing an architecture sensitive to high-frequency details, followed by fine-tuning through a LoRA (Low-Rank Adaptation) module guided by high-frequency information.

**Questions:**

- In Section 3, the authors conduct extensive analysis of the VAE. What is the configuration of the VAE with stronger reconstruction capability? What are the design and training differences between the two baseline models? Can it be proven that low-frequency or high-frequency information in the low-dimensional latent space is strongly correlated with the low/high-frequency information in the corresponding model results in the pixel domain?
- Since this paper aims to solve high-resolution image restoration (e.g., UHD resolution), why are the metric calculations in Tables 1 and 2 only applicable to 256×256 resolution?
- Different restoration tasks have varying dependencies on low/high frequencies, but the method's design still focuses on recovering high-frequency information. Intuitively, its generalization to multi-type degradation tasks relies on Equation 3, which is equivalent to moving the so-called "second-order degradation" from the image domain to the latent space. Has the method made specific designs for low-frequency selection or frequency adaptation, or analyzed the effectiveness of each proposed stage for each task?

**Ethical Concerns:**

["NO or VERY MINOR ethics concerns only"]

**Final Justification:**

Please see the comment.

**Limitations:**

The limitation of proposed method is discussed in the supplementary material.

**Paper Formatting Concerns:**

There is no formatting concern.

**Quality:**

3

**Strengths And Weaknesses:**

Strength
- The proposed method has a relatively lightweight computational complexity and parameter count compared to similar approaches, making it suitable for handling high-resolution content.
- The proposed modules enable the inpainting model to adapt to various types of degradation.
- Experiments show that the method can achieve superior results in metrics such as PSNR and SSIM.

Weakness
- Figure 4 is best accompanied by a comparison of more detailed regions, such as local textures.
- It is recommended to give a separate comparison for the restoration of text-type content in Figure 4.

---

> ### Author Rebuttal · Authors · 2025-07-30
>
> ### W1 & W2: Fig 4 Layout
> We will revise Fig 4 in the revision to enhance the visual comparison of regions and text, ensuring clearer presentation of the results.
>
> ### Q1: VAE Baseline Configurations and Frequency Correlation Between Latent and Pixel Domains
>
> 1. Detailed VAE Baseline Configurations
>
>     The configurations of the VAE baselines are detailed in Section B.2 ("Baseline Construction") of the Supplementary Material. For clarity, we summarize the key distinctions below:
>
>     VAE₁ (Stronger Generalization): Designed to encode robust semantic information, this model uses a KL loss weight of ($\lambda_{KL} = 1 \times 10^{-4}$), a downsampling factor of 32, and 8 latent channels. The higher KL weight and compression rate promote smoother, semantics-focused latent representations.
>
>     VAE₂ (Stronger Reconstruction): Optimized for comprehensive image reconstruction, this model employs a KL loss weight of ($\lambda_{KL} = 1 \times 10^{-8}$), a downsampling factor of 8, and 32 latent channels. The minimal KL weight and lower compression rate enable retention of high-frequency details for precise reconstruction.
>
> 2. Frequency Correlation Between Latent and Pixel Domains
>
>     * The relationship between the spectral distribution of the VAE latent space and the decoded image has been discussed in detail in [1, 2]. The latent space’s spectral distribution directly influences the frequency distribution of the output image, though this relationship is non-linear. As noted in [1], the VAE decoder exhibits a spectral bias that attenuates high-frequency components, as reconstruction-oriented VAEs tend to encode excessive high-frequency information, introducing noise. This is evidenced by the latent space visualization of our baseline model VAE₂, designed for enhanced reconstruction, in Supplementary **Figure 5**. Experimental results in **Figure 2(c)** further demonstrate that VAE₂ encodes a significantly higher proportion of high-frequency components in its latent space compared to the generalization-focused  VAE₁, which exhibits lower high-frequency content. This clearly indicates a direct correlation between the latent space’s frequency composition and the output image’s characteristics (e.g., high-fidelity, detail-rich features). To illustrate this relationship, we provide PSNR metrics for the low-frequency (LF) and high-frequency (HF) components of the reconstruction outputs for both baselines, as shown below. The comparable LF-PSNR and primary differences in HF detail recovery confirm the **direct link between latent space frequency content and reconstruction quality**.
>
>    * Consequently, our findings reveal that **effective high-frequency information** in latent space promotes richer high-frequency details in reconstructed images, but a significant portion of this information constitutes **invalid noise**. To address this, our proposed equivariance loss $L_{Eqv}$ balances the latent space’s spectral distribution, reducing reliance on excessive high-frequency signals and indirectly filtering out invalid high-frequency components. This effect is demonstrated in the fifth column of **Supplementary Figure 5**, while **Supplementary Fig6**’s high-frequency truncation experiment further confirms the robustness of $L_{Eqv}$ in mitigating high-frequency attenuation.
>
>     | Baseline | PSNR(LF) | PSNR(HF)  |PSNR |
>     | - | - | - | - |
>     | $VAE_{1}$ |39.4|18.2| 24.3|
>     | $VAE_{2}$ |39.8|32.9|36.4|
>
> [1] Simpler is Better: Spectral Regularization and Up-Sampling Techniques for Variational Autoencoders [ICASSP 2022]
>
> [2] FreqMark: Invisible Image Watermarking via Frequency Based Optimization in Latent Space[NIPS 2024]
>
> ---
>
> ### Q2: Image Resolution for Metrics Computation in Tables
> The FLOPs reported in Tables 1 and 2 were calculated at a standard resolution of 256×256 to ensure a fair comparison of computational complexity with baseline methods, a common practice in image restoration. Many state-of-the-art (SOTA) methods cannot process full 4K images on consumer-grade GPUs, necessitating a unified, smaller resolution to evaluate inherent model efficiency, consistent with UHDProcessor metrics. To further demonstrate our method’s performance, we conducted additional tests on a high-memory GPU (A100 80GB) at a larger resolution (1024×1024). The results, presented below, highlight our method’s pronounced **efficiency advantage in higher-resolution scenarios**.
>
> | Method   | PromptIR | Histoformer | Gridformer | AdaIR | UHDprocessor | Ours |
> | -------- | -------- | ----------- | ---------- | --------- | ------------ | ---- |
> | FLOPS(G) | 2530     | 1465        | 5878       | 2354      | 89           | 59   |
>
> ---
>
> ### Q3: Frequency Adaptability and Task-Specific Analysis
>
> The reviewer accurately notes that different tasks exhibit varying dependencies on frequency components. Our framework addresses this through the synergistic interplay of two stages, rather than solely relying on high-frequency restoration, ensuring a degradation-agnostic approach with robust generalization.
>
> * **Stage 1 - Universal Foundation:** Stage 1 is designed to construct a robust, frequency-balanced latent space foundation that is not limited to high-frequency components.
>
>     * Low-Frequency Semantic Preservation: The Progressive Degradation-aware Pretraining Strategy (PDPS, Eq. 3) and the degradation-invariant visual semantic loss ((L_{Inv})) prioritize preserving low-frequency semantic structures, enabling the model to capture core semantic information agnostic to degradation types. This generalization capability stems not only from the multi-degradation adaptability of PDPS (Eq. 3) but also from its synergy with (L_{Inv}), which explicitly enhances degradation-invariant encoding in the VAE.
>     * Frequency Balance: The latent equivariance loss ((L_{Eqv})) promotes consistency across different scales, reducing over-reliance on specific frequency components and fostering a balanced frequency representation in the latent space.
>
> * **Stage 2 - Controlled Fine-Tuning:** The focus on high-frequency restoration in this stage addresses the inherent loss of high-frequency information during VAE encoding, rather than task-specific high-frequency dependencies. To preserve the robust, balanced latent space from Stage 1, we introduce HF-LoRA for fine-tuning. This approach enhances the extraction of high-frequency details critical for restoration tasks while maintaining the integrity of the latent structure.
>
> Although different degradations emphasize distinct frequency bands for restoration, our method’s Stage 1 leverages **degradation invariance** and **frequency balance** to enable the VAE to extract **semantically relevant information** across all degradations. Consequently, our approach is **degradation-agnostic**, requiring no specific designs for particular degradation types, as evidenced by the significant performance improvements in **Tables 1 and 2**. Furthermore, this degradation-agnostic design enhances **generalization, with our method demonstrating substantial advantages in unseen and composite degradation scenario**s, as shown in **Table 4** of the main paper.

---

> > ### Comment · Reviewer_KgHd · 2025-08-07
> >
> > After reading the authors’ response, I appreciate that most of my concerns have been addressed. However, regarding Q3, the explanation suggests that frequency adaptability operates as a multi-stage refinement process. In this case, the degradation observed in the initial stage may be mitigated in subsequent refinement stages. To support the claim, I recommend including an ablation study that explicitly demonstrates the contribution of each stage to the final performance. Nevertheless, I acknowledge the innovative aspects of the proposed method and the overall quality of the work. Therefore, I have decided to keep my original rating unchanged.

---

> > > ### Author Response · Authors · 2025-08-08
> > > **Reply to Reviewer KgHd**
> > >
> > > We sincerely thank the reviewer for their valuable feedback and for recognizing the innovation and overall quality of our work. We fully agree that quantifying the contribution of each stage through a clear ablation study is crucial. The stage-wise contribution analysis you suggested can be found in our main paper's **Table 5(a) "Alation Studies"**, where the contribution of each stage is clearly presented:
> > >
> > > - **Contribution of Stage 1:** Building a General and Robust Restoration Foundation
> > >
> > >     The result in the "w/o Fine-Tuning" row (PSNR 28.48 dB) represents the performance of our restoration task built upon the universal latent space foundation constructed in Stage 1. Furthermore:
> > >     - Removing the **semantic alignment loss** $L_{Inv}$ ("w/o $L_{Inv}$") causes a sharp performance drop of **4.2 dB**, highlighting the significant impact of latent space semantics on the final result.
> > >     - Removing the **latent equivariance loss** $L_{Eqv}$ ("w/o $L_{Eqv}$") leads to a substantial drop of **2.8 dB**, demonstrating the importance of frequency balance within the latent space.
> > >     - Removing the **progressive perturbation PDPS** ("w/o PDPS") results in a **0.66 dB** performance decrease, proving the contribution of perturbation robustness in the latent space.
> > >
> > >     These results indicate that **the degradation-agnostic and frequency-balanced latent space**, established through progressive degradation perturbation, semantic invariance, and decoding equivariance constraints, serves as a critical foundation for adapting to different degradations and frequencies, significantly impacting the final performance.
> > >
> > > - **Contribution of Stage 2:** Achieving Precise and Effective Performance Refinement
> > >
> > >     The result in the "Latent Harmony" row (PSNR 29.77 dB) represents the final performance of our complete two-stage synergistic framework.
> > >     - Compared to the Stage 1 baseline (28.48 dB), this marks a significant PSNR improvement of **+1.29 dB**.
> > >     - This demonstrates that the **high-frequency refinement** in Stage 2, while preserving the robust and balanced latent space from Stage 1, can precisely compensate for the inherent high-frequency loss of VAE encoding based on the restoration objective. This achieves a superior synergistic effect with Stage 1 and overcomes its performance bottleneck.
> > >
> > > - **Conclusion**：Therefore, the detailed ablation results for each component provide strong evidence of the impact of each stage on the final outcome, as well as the synergistic effect between them. The interplay of these two stages renders our entire framework **degradation-agnostic**, eliminating the need for specific designs for particular degradation types. This leads to **stronger degradation adaptability** (as shown in Tables 1 and 2) and **superior generalization** to unseen degradations (as shown in Table 4).
> > >
> > > We will revise the presentation of the ablation study in our revision to more clearly present the contribution of each stage to the final performance. We thank the reviewer for their insightful feedback. Should any remaining questions or uncertainties arise, we would be happy to provide further clarification.

---

### Official Review · Reviewer_tSuv · 2025-07-02

**Clarity:** 2
**Significance:** 2
**Originality:** 2
**Rating:** 4
**Confidence:** 4

**Summary:**

This paper introduces Latent Harmony, a two-stage unified framework for UHD image restoration that addresses key limitations of Variational Autoencoders (VAEs) in balancing semantic generalization and high-frequency detail reconstruction. Stage 1 develops a regularized VAE (LH-VAE) using progressive degradation perturbation, semantic alignment (via DINOv2), and latent equivariance to learn a robust and frequency-balanced latent space. Stage 2 applies a novel high-frequency-guided LoRA (HF-LoRA) mechanism, separating fidelity- and perception-oriented fine-tuning paths for the encoder and decoder, respectively.

**Questions:**

1. While the two-stage design is central, the interplay between the Stage 1 VAE latent structure and Stage 2 LoRA tuning (especially how much the LoRA fine-tuning modifies or respects the original latent semantics) is not clearly quantified.  Please provide quantitative or visual analysis (e.g., latent feature similarity, latent traversals) to show how the LoRA updates preserve or adapt the semantic structure from Stage 2.

2. In Section 5.1, the authors claim that their method "effectively removes degradations while preserving intricate background textures." However, the qualitative results in Figure 4 do not convincingly support this claim. In fact: The visual improvements over other baselines (e.g., HAIR, UHDprocesser) appear marginal at best;  Some restored outputs from your method still exhibit blurring, noise, or residual artifacts, especially in complex background regions;

3.In Tables 1 and 2, only one other UHD-capable method (UHDprocesser ) is included for direct 4K full-size evaluation. However, recent works such as DreamUHD, and ERR have demonstrated UHD restoration capabilities, though perhaps indirectly or under modified setups.

Liu, Yidi, et al. "DreamUHD: Frequency Enhanced Variational Autoencoder for Ultra-High-Definition Image Restoration." Proceedings of the AAAI Conference on Artificial Intelligence. Vol. 39. No. 6. 2025.
Zhao, Chen, et al. "From Zero to Detail: Deconstructing Ultra-High-Definition Image Restoration from Progressive Spectral Perspective." Proceedings of the Computer Vision and Pattern Recognition Conference. 2025.

4.Could you explain how Param and FLOPs are decreased in Table 5(c)?

**Ethical Concerns:**

["NO or VERY MINOR ethics concerns only"]

**Final Justification:**

While the main components, such as VAE regularization, LoRA fine-tuning, and loss design, are based on existing paradigms, I appreciate the significant effort made to adapt these for UHD image restoration. The authors have also clarified minor errors in the figures and addressed issues in the manuscript during the rebuttal phase, providing additional experiments for further validation. Based on their response, I will raise my overall evaluation.

**Limitations:**

yes

**Quality:**

3

**Strengths And Weaknesses:**

Quality: This work presents a  VAE-based framework for UHD image restoration. The authors first justify their motivation through statistical analysis, then detail the loss function and network architecture. Extensive experiments validate each component, demonstrating a complete methodology.

Clarity: UHD image restoration combined with VAEs is natural and well-motivated, and the paper presents a clearly structured framework. To enhance readability, the dimensions of certain figures (e.g., Figures 1 and 4) could be further optimized.

Significance: The study constitutes a valuable engineering investigation of VAE-based approaches, although many design elements (Latent Space Representation and LoRA) follow an incremental improvement paradigm.

Originality:  Its novelty remains limited as it fails to provide new fundamental insights for UHD image restoration. The observed generalization capabilities appear to derive primarily from the inherent properties of VAEs.

---

> ### Author Rebuttal · Authors · 2025-07-30
>
> ### W1: Figure Size
> We will revise the size and layout of our figures in the revision to improve readability.
>
> ### W2: Insight and Novelty
> We thank the reviewer for evaluating our work as a "valuable engineering investigation." However, we wish to clarify that our core contribution lies not in inventing isolated technical components, but in providing a new analytical perspective and a systematic solution for the complex task of UHD All-in-One (UHD AIO) image restoration. We believe this goes beyond "incremental improvement."
>
> **Our Core Insight:** Breaking the UHD AIO Performance Bottleneck from the Perspective of VAE Latent Space Optimization.
>
> The complexity of the UHD AIO task stems from the need to simultaneously address the demand for computational efficiency due to ultra-high resolutions and the challenge of joint optimization across multiple degradation types. Traditional methods typically devise separate solutions for these problems, but the combination of such isolated designs often leads to compatibility issues and suboptimal performance (i.e., the whole is less than the sum of its parts).
>
> Our work is the first to unify these problems from the perspective of VAE latent space optimization. Through extensive observational experiments presented in **Fig 2**, we have distilled the challenge into a new analytical framework centered on the trade-offs among three core elements:
>
> 1. **At the Latent Space Level:** The trade-off between content "Generalization" and detail "Reconstruction Capability."
>
> 2. **At the Optimization Level:** The conflict between "Fine-tuning Adaptation" for downstream tasks and "Structural Preservation" of the VAE prior.
>
> 3. **At the Restoration Result Level:**  The competition between the "Fidelity" and "Perceptual Quality" of the decoded image.
>
> To our knowledge, this is the first work in the UHD AIO field to **unify the task's multiple challenges into the perspective of latent space optimization** and to further deconstruct them systematically into **three core trade-offs** through experimental analysis. This points to a **new analytical perspective and optimization direction** for the field, which we consider a key insight. Based on this, the Latent Harmony framework we propose is not a simple combination of techniques, but a systematic solution precisely designed to address the aforementioned trade-offs.
>
> **Our Innovative Solution:** A Synergistic Framework for Resolving a Triad of Trade-offs
> 1. Addressing the "Generalization vs. Reconstruction" Conflict: **Objective-driven VAE Regularization (LH-VAE)**
>
>      Our work transcends the mere application of a standard VAE. Through comprehensive analysis in Section 3.1 (**Fig 2a,b,c**), we introduce a novel frequency encoding perspective that reveals the inherent conflict in standard VAEs: the trade-off between latent space “generalization” and “reconstruction” capabilities for the UHD AIO task. Leveraging this insight, we propose the Latent Harmony VAE (LH-VAE), which employs a joint regularization strategy integrating the Progressive Degradation-aware Pretraining Strategy (PDPS), semantic alignment loss $L_{Inv}$, and latent equivariance loss $L_{Eqv}$ . This objective-driven design reshapes the VAE's latent space to simultaneously meet the dual demands for robust generalization and detailed reconstruction, harnessing the VAE's intrinsic properties for the specific challenges of UHD AIO task.
>
> 2. Addressing the "Adaptation vs. Preservation" Conflict: **Decoupled Co-optimization via a High-Frequency "Bridge"**
>
>     To address the inherent conflict between “performance adaptability” and “structural preservation” when fine-tuning VAEs for downstream tasks (see **Fig 2d**), we propose an innovative decoupled collaborative optimization strategy.This approach leverages a high-frequency alignment loss as a “bridge” to separate the VAE’s optimization from the gradient flow of the primary restoration loss. It preserves the VAE's generalization capability by protecting the stable, low-frequency semantic structure, while simultaneously adapting to the downstream task by precisely compensating for the suppressed high-frequency details. This strategy enables efficient and stable adaptation of the VAE, breaking through the performance bottlenecks of the first stage.
>
> 3. Addressing the "Perception vs. Fidelity" Conflict: **An Objective-Decoupling LoRA Fine-tuning Paradigm (HF-LORA)**
>
>     Drawing on the differential fine-tuning effects observed in the VAE encoder and decoder (**Fig 2e**), we architecturally decouple the conflicting objectives of perceptual quality and fidelity:
>
>     * Encoder for “Extraction”: We employ a Fidelity-oriented HF-LORA (FHF-LORA), which focuses on faithfully extracting traceable high-frequency structures from the input .
>     * Decoder for “Generation”: We use a Perception-oriented HF-LORA (PHF-LORA), which excels at generating visually plausible and sharp high-frequency textures .
>     * Collaboration and Control: Alternating optimization mitigates gradient conflicts, while an adjustable parameter α provides flexible trade-off control during inference, empowering users to tailor outputs to specific needs.
>
> Our work delivers a dual contribution to the UHD AIO task. First, it reframes the task’s core challenges by systematically distilling them into three fundamental trade-offs within latent space representation optimization. Second, our proposed Latent Harmony framework is not a collection of isolated components but a synergistic system meticulously designed to address these trade-offs. This framework effectively tackles challenges at each level while achieving a remarkable synergistic effect, significantly enhancing the overall performance of UHD AIO. Consequently, we believe our work offers a robust and novel contribution to the field, both through the depth of its problem analysis and the ingenuity of its framework design.
>
> ###  Q1: Quantitative and Visual Analysis of HF-LoRA Fine-Tuning
> **Visualization Analysis:** The visualization results, as shown in **Supp Fig 5** (columns 5 and 6), provide a comparative analysis of the latent representations before and after HF-LoRA fine-tuning. These results demonstrate that post-fine-tuning, the fundamental semantic and structural information is preserved, while significantly enhancing high-frequency texture details.
>
> **Quantitative Metrics:** We quantitatively evaluated the interplay between Stage 1 and Stage 2 latents by computing the Cross-Dataset Consistency Score (CDCS) and the high-frequency energy ratio (HFER). The experimental results indicate that after HF-LoRA fine-tuning, CDCS exhibits only a marginal decrease, while the high-frequency energy ratio shows a substantial increase. These findings quantitatively confirm that HF-LoRA effectively enhances high-frequency information beneficial to restoration outcomes while maintaining the integrity of semantic information.
>
> |        | CDCS | HFER |
> | - | - | - |
> | stage1 | 8.4  | 15%  |
> | stage2 | 8.1  | 23%  |
>
> ###  Q2: Fig 4 Visual Results
>
> We thank the reviewer for their insightful evaluation of Fig 4’s visual results and valuable feedback. While some improvements may seem subtle in small-scale figures, we highlight key advancements, supported by quantitative metrics and user studies, to demonstrate our method’s effectiveness.
>
> Task-Specific Advantages: In low-light enhancement, our method restores sharper background text details, unlike blurrier outputs from competing methods. For dehazing, it more effectively removes haze, recovering authentic contours and colors of distant buildings. In deblurring, it enhances human outlines with reduced ringing artifacts. For denoising, we apologize for an error in Fig 4, where the “Ours” result was mistakenly replaced with UHD-Processor’s, causing identical outputs. Corrected results, verified internally, will be updated in the revised manuscript to showcase superior facial detail restoration. Comprehensive comparisons, including denoising, are in **Supp Fig 1 and 2**.
>
> Quantitative Superiority: Tables 1 and 2 confirm our method’s consistent outperformance across all degradation tasks in PSNR, SSIM, and LPIPS.
>
> Subjective Perceptual Advantages: A user study with 20 volunteers (**Supp Fig 3**) shows our restored images received significantly higher subjective ratings than baselines.
>
> In summary, our method achieves clear advancements in degradation removal, texture preservation, and human perception alignment, validated by visual comparisons, quantitative metrics, and user studies. We will include higher-quality figures with magnified details in the revised manuscript to better highlight these strengths.
>
> ### Q3: More UHD Baseline Comparisons
>
> We have supplemented our results with comparisons to DreamUHD and ERR, as presented below. Since these methods are designed for single-task UHD image restoration, their performance is suboptimal in the All-in-One scenario. This underscores the necessity of tailored designs for the UHD All-in-One task.
>
> |Method|DreamUHD|ERR|Ours|
> |-|-|-|-|
> |PSNR|27.2|27.8|29.7|
> |SSIM|0.853|0.836|0.877|
>
> ### Q4: Reasons for Parameter and FLOPs Reduction in Tab 5c
>
> The substantial reduction in parameters and FLOPs in Tab 5c arises from our VAE framework’s key strength: shifting computations of heavy restoration networks (e.g., Restormer, NAFNet, SFHformer) from pixel space to an efficient latent space. The process is as follows:
>
> * Encode: Our lightweight LH-VAE encoder compresses high-resolution images into compact latent representations.
> * Latent Restoration: The restoration network processes these low-dimensional latents, significantly reducing computational complexity tied to input size and channels.
> * Decode: The LH-VAE decoder reconstructs the restored latents into high-resolution images.
>
> Thus, the “+Ours” metrics reflect the combined cost of LH-VAE and the latent-space restoration network, achieving 80%–95% efficiency gains while preserving performance.

---

> > ### Comment · Reviewer_tSuv · 2025-08-04
> >
> > Thank the authors for addressing my concerns. While the main components, such as VAE regularization, LoRA fine-tuning, and loss design are based on existing paradigms, I appreciate the significant effort made to adapt these for UHD image restoration. Given the authors’ detailed response, I will raise my overall evaluation.

---

> > > ### Author Response · Authors · 2025-08-04
> > > **Reply to Reviewer tSuv**
> > >
> > > We sincerely appreciate the reviewer's thoughtful feedback and recognition of our rebuttal efforts. We will rigorously implement these suggestions to further enhance.

---

### Note · Authors · 2025-08-12

We appreciate the reviewers' initial positive feedback on our work's motivation (tSuv, KgHd, kppD, Nq1u), novelty (KgHd, Nq1u), comprehensive experiments (KgHd, kppD, Nq1u), efficiency (kppD, KgHd), and SOTA results with strong generalization (KgHd, kppD).

Our primary contribution is the introduction of a new analytical perspective for the UHD All-in-One task using VAE. We were the first to systematically diagnose the performance bottleneck as a series of **three fundamental trade-offs** within the VAE latent space (Generalization vs. Reconstruction, Adaptation vs. Preservation, and Fidelity vs. Perception). Consequently, our proposed Latent Harmony framework is not a collection of isolated techniques, but a **synergistic system** meticulously designed to resolve these specifically identified trade-offs.

During the rebuttal phase, we were pleased that our detailed responses and additional quantitative results—including new baseline comparisons, further ablations on different backbones and PEFT methods, and clarifications on our stage-wise contributions from existing tables—were well-received.
* **Reviewer tSuv** explicitly stated that the concerns were addressed, acknowledged our contribution to the UHD field, and committed to **raising the score**.
* **Reviewer KgHd** indicated that most of their concerns were resolved and **maintained positive rating**. We subsequently clarified the stage-wise contribution analysis as requested.
* **Reviewer kppD** confirmed that all concerns were addressed, expressed hope for a future code release to benefit the community, and committed to **raising the score**.
* **Reviewer Nq1u**  gave a **positive initial rating**, raised the scores for Quality, Clarity, and Originality, and increased the Confidence after our detailed responses.

In our revision, we will enhance the visual quality and informational clarity of our figs and tabs, add key methodological clarifications, and integrate all new experiments from the rebuttal phase.  Furthermore, we commit to **open-sourcing our code** upon acceptance of the paper to foster community development.

We sincerely thank all the reviewers for their constructive feedback and for their recognition of our contributions. The reviewers' insightful suggestions are crucial for improving the quality, clarity, and impact of our paper. We would also like to express our gratitude to the reviewers and the AC for their valuable time and effort throughout this process.

---

### Decision · Program_Chairs · 2025-09-17

**Decision:**

Accept (poster)

**Comment:**

The paper presents a robust and well-thought-out framework that tackles a challenging problem in image restoration. The authors' systematic approach to identifying and solving trade-offs within the VAE latent space, combined with their effective rebuttal, has convinced the reviewers of the paper's merit. The AC-panel concurs with their decision and recommends acceptance.